# SegmentMeIfYouCan:
# A Benchmark for Anomaly Segmentation

**Robin Chan**[*1] **Krzysztof Lis**[*2] **Svenja Uhlemeyer**[*1] **Hermann Blum**[*3] **Sina Honari**[2]

**Roland Siegwart**[3] **Pascal Fua**[2] **Mathieu Salzmann**[2] **Matthias Rottmann**[1]

## Abstract

State-of-the-art semantic or instance segmentation deep neural networks (DNNs) are usually trained on a closed set of semantic classes. As such, they are ill-equipped to handle previously-unseen objects. However, detecting and localizing such objects is crucial for safety-critical applications such as perception for automated driving, especially if they appear on the road ahead. While some methods have tackled the tasks of anomalous or out-of-distribution object segmentation, progress remains slow, in large part due to the lack of solid benchmarks; existing datasets either consist of synthetic data, or suffer from label inconsistencies. In this paper, we bridge this gap by introducing the "SegmentMeIfYouCan" benchmark. Our benchmark addresses two tasks: Anomalous object segmentation, which considers any previously-unseen object category; and road obstacle segmentation, which focuses on any object on the road, may it be known or unknown. We provide two corresponding datasets together with a test suite performing an in-depth method analysis, considering both established pixel-wise performance metrics and recent component-wise ones, which are insensitive to object sizes. We empirically evaluate multiple state-of-the-art baseline methods, including several models specifically designed for anomaly / obstacle segmentation, on our datasets and on public ones, using our test suite. The anomaly and obstacle segmentation results show that our datasets contribute to the diversity and difficulty of both data landscapes.

## 1 Introduction

The advent of high-quality publicly-available datasets, such as Cityscapes [1], BDD100k [2], A2D2 [3] and COCO [4] has hugely contributed to the progress in semantic segmentation. However, while state-of-the-art deep neural networks (DNNs) yield outstanding performance on these datasets, they typically provide predictions for a closed set of semantic classes. Consequently, they are unable to classify an object as *none of the known categories* [5]. Instead, they tend to be overconfident in their predictions, even in the presence of previously-unseen objects [6], which precludes the use of uncertainty to identify the corresponding anomalous regions.

Nevertheless, reliability in the presence of unknown objects is key to the success of applications that have to face the diversity of the real world, *e.g.*, perception in automated driving. This has motivated the creation of benchmarks such as Fishyscapes [7] or CAOS [8]. While these benchmarks have enabled interesting experiments, the limited real-world diversity in Fishyscapes, the lack of a

---

[*]equal contribution

[1]Stochastics Group, IZMD, University of Wuppertal, Wuppertal, Germany

[2]Computer Vision Laboratory, EPFL, Lausanne, Switzerland

[3]Autonomous Systems Lab, ETH, Zürich, Switzerland

35th Conference on Neural Information Processing Systems (NeurIPS 2021) Track on Datasets and Benchmarks.

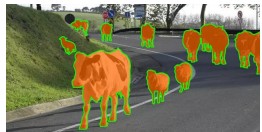 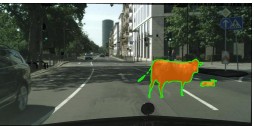 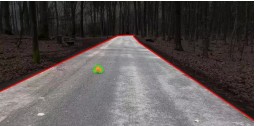 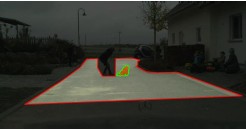

| Ours: RoadAnomaly21 | Fishyscapes | Ours: RoadObstacle21 | LostAndFound |

Figure 1: Comparison of images from our and existing public datasets. Anomalies / obstacles are highlighted in orange, darkened regions are excluded from the evaluation. In RoadAnomaly21, anomalies may appear everywhere in the image. In contrast to Fishyscapes, where anomalous objects are synthetic, all RoadAnomaly21 images are real. In RoadObstacle21, the region of interest is restricted to the drivable area with obstacles ahead. This is comparable to LostAndFound, where the labeling, however, is not always consistent, *e.g.* children are anomalies but other humans not.

public leader board and of a benchmark suite in CAOS, and the reliance on synthetic images in both benchmarks hinder proper evaluation of and comparisons between the state-of-the-art methods.

In this paper, motivated by the limitations of existing anomaly segmentation datasets and by the emerging body of works in this direction [7, 9, 10, 11, 12, 13, 14, 15, 16], we introduce the *SegmentMeIfYouCan*[2] benchmark. It is accompanied with two datasets, consisting of diverse and manually annotated real images, a public leader board and an evaluation suite, providing in-depth analysis and comparisons, to facilitate the development of road anomaly segmentation methods.

Our benchmark encompasses two separate tasks. The first one consists of strict anomaly segmentation, where any previously-unseen object is considered as an anomaly. Furthermore, motivated by the observation that the boundary between known and unknown classes can sometimes be fuzzy, for instance for *car vs. van*, we introduce the task of obstacle segmentation, whose goal is to identify all objects on the road, may they be from known classes or from unknown ones.

For the anomaly track, we provide a dataset of 100 images with pixel-wise annotations over two classes (anomaly, not anomaly) and a void class, which, in analogy to Cityscapes, signals the pixels that are excluded from the evaluation. We consider any object that strictly cannot be seen in the Cityscapes data as anomalous, appearing anywhere in the image. For the obstacle track, our dataset contains 327 images with analogous annotation (obstacle, not obstacle, void), and focuses only on the road as region of interest. The focus in this track is of more practical need, *e.g.* for automated driving systems, targeting obstacles that may cause hazardous street situations, see Figure 1. All images of our datasets are publicly available for download[2], together with a benchmark suite that computes both established pixel-wise metrics and recent component-wise ones.

In the remainder of this paper, we first review existing anomaly detection datasets, methods and evaluation metrics in more detail. We then describe our new benchmark and provide extensive experiments comparing state-of-the-art road anomaly / obstacle segmentation methods on our datasets and on other related ones, showing the difficulty of the models on the proposed benchmarks.

## 2 Related Work

In this section we first review previous datasets for anomaly detection, with some of them being designed for road anomaly segmentation. Then we briefly describe some of the methods on anomaly and obstacle segmentation.

### 2.1 Datasets and Benchmarks

Existing methods for anomaly detection have often been evaluated on their ability to separate images from two different source distributions, such as separating MNIST from FashionMNIST [17, 18, 19], NotMNIST [19], or Omniglot [20], and separating CIFAR-10 from SVHN [18, 19, 21] or LSUN [18, 21, 22]. Such experiments can be found in many works, including [17, 18, 19, 21, 22, 23].

---

[2]https://www.segmentmeifyoucan.com/

| Dataset | anomaly pixels | non-anomaly pixels | diverse scenes | different anomalies | dataset size | ground truth (gt) components | mean & std of gt size relative to image size |
|---|---|---|---|---|---|---|---|
| Fishyscapes LostAndFound val [7] | 0.23% | 81.13% | 12 | 7 | 373 | 165 | 0.13% ± 0.23% |
| CAOS BDD-Anomaly test [8] | 0.83% | 81.28% | 810 | 3 | 810 | 1231 | 0.55% ± 1.84% |
| Ours: RoadAnomaly21 test | 13.83% | 82.17% | 100 | 26 | 100 | 262 | 4.12% ± 7.29% |
| LostAndFound test (NoKnown) [25] | 0.12% | 15.31% | 13 (12) | 9 (7) | 1203 (1043) | 1864 (1709) | 0.08% ± 0.16% |
| LiDAR guided Small Obstacle test [26] | 0.07% | 36.09% | 2 | 6 | 491 | 1203 | 0.03% ± 0.07% |
| Ours: RoadObstacle21 test | 0.12% | 39.08% | 8 | 31 | 327 | 388 | 0.10% ± 0.25% |

| Dataset (as above) | labels are private | weather conditions | geography |
|---|---|---|---|
| Fishyscapes val | (✓in test set) | clear | DE |
| CAOS BDD test | ✗ | clear, snow, night, rain | US |
| Ours: RA21 test | ✓ | clear, snow | global |
| LaF test | ✗ | clear | DE |
| Small Obs. test | ✗ | clear | IN |
| Ours: RO21 test | ✓ | clear, snow, night | CH, DE |

Table 1: Main properties of comparable real-world anomaly (top three rows) and obstacle (bottom three rows) segmentation datasets. Our main contribution is the diversity of the anomaly (or obstacle) categories and of the scenes. Note that "void" pixels are not included in this table.

For semantic segmentation, a similar task was therefore proposed by the WildDash benchmark [24] that analyzes semantic segmentation methods trained for driving scenes on a range of failure sources, including full-image anomalies, such as images from the beach. In our work, by contrast, we focus on the problem of robustness to anomalies that only cover a small portion of the image, and on the methods that aim to segment such anomalies, *i.e.* method for the task of *anomaly segmentation*.

One prominent dataset tackling the task of anomaly segmentation is LostAndFound [25], which shares the same setup as Cityscapes [1] but includes anomalous objects / obstacles in various street scenes in Germany. LostAndFound contains 9 different object types as anomalies, and only has annotations for the anomaly and the road surface. Furthermore, it considers children and bicycles as anomalies, even though they are part of the Cityscapes training set, and it contains several labeling mistakes. Although we filter and refine LostAndFound in this work[3], similar to Fishyscapes [7], the low diversity of anomalies persists.

The CAOS BDD-Anomaly benchmark [8] suffers from a similar low-diversity issue, arising from its use of only 3 object classes sourced from the BDD100k dataset [2] as anomalies (besides including several labeling mistakes, see Appendix F.5). Both Fishyscapes and CAOS try to mitigate this low diversity by complementing their real images with synthetic data. Such synthetic data, however, is not realistic and not representative of the situations that can arise in the real world.

In general, the above works illustrate the shortage of diverse real-world data for anomaly segmentation. Additional efforts in this regard have been made by sourcing and annotating images of animals in street scenes [14], and by leveraging multiple sensors, including mainly LiDAR, to detect obstacles on the road [26]. In any event, most of the above datasets are fully published with annotations, allowing methods to overfit to the available anomalies. Furthermore, apart from Fishyscapes, we did not find any public leader boards that allow for a trustworthy comparison of new methods. To provide a more reliable test setup, we do not share the labels and request predictions of the shared images to be submitted to our servers. Furthermore, we provide a leader board, which we publish alongside two novel real-world datasets, namely RoadAnomaly21 and RoadObstacle21. A summary of the main properties of the mentioned datasets is given in Table 1. Our main contribution in both proposed datasets is the diversity of the anomaly categories and of the scenes.

In RoadAnomaly21, anomalies can appear anywhere in the image, which is comparable to Fishyscapes LostAndFound [7] and CAOS BDD-Anomaly [8]. Although the latter two datasets are larger, their images only show a limited diversity of anomaly types and scenes because they are usually frames of videos captured in single scenes. By contrast, in our dataset every image shows a unique scene, with at least one out of 26 different types of anomalous objects and each sample widely differs in size, ranging from 0.5% to 40% of the image.

In RoadObstacle21, all anomalies (or obstacles) appear on the road, making this dataset comparable to LostAndFound [25] and the LiDAR guided Small Obstacle dataset [26]. Again, the latter two datasets contain more images than ours, however, the high numbers of images result from densely sampling frames from videos. Consequently, those two datasets lack in object diversity (9 and 6 categories, respectively, compared to 31 in our dataset). Furthermore, the videos are recorded under perfect weather conditions, while RoadObstacle21 shows scenes in diverse situations, including night, dirty roads and snowy conditions.

---

[3]In the following, we refer to the LostAndFound subset without the images of children, bicycles and invalid annotations as "LostAndFound-NoKnown".

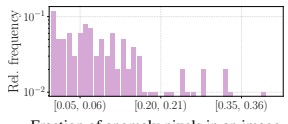 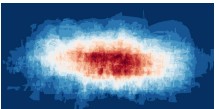 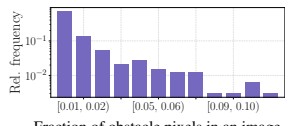 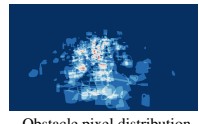

|   | |   | |
|---|---|---|---|
| (a) RoadAnomaly21 | | (b) RoadObstacle21 | |

Figure 2: Relative frequency of annotated anomaly / obstacle pixels within an image over the 100 images in the RoadAnomaly21 test dataset (left) and the 327 images in the RoadObstacle21 test dataset (right), respectively. Here, the fraction of anomaly / obstacle pixels serves as a proxy for the size of the objects of interest within an image. Note that the y-axes of the histograms are $\log$ scaled.

## 2.2 Anomaly and Obstacle Segmentation

Anomaly detection was initially tackled in the context of image classification, by developing post-processing techniques aiming to adjust the confidence values produced by a classification DNN [6, 18, 21, 22, 23]. Although originally designed for image-level anomaly detection, most of these methods can easily be adapted to anomaly segmentation [7, 9] by treating each individual pixel in an image as a potential anomaly.

Another relevant approach consists of estimating the uncertainty of the predictions, leveraging the intuition that anomalous image regions should correlate with high uncertainty. One way of doing so is Bayesian (deep) learning [27, 28], where the model parameters are treated as distributions. Because of the computational complexity, approximations to Bayesian inference have been developed [29, 30, 31, 32] and extended to semantic segmentation [33, 34, 35]. Instead of reasoning about uncertainty, other non-Bayesian approaches tune previously-trained models to the task of anomaly detection by either modifying its architecture or exploiting additional data. For example, in [36], anomaly scores are learned by adding a separate branch to the DNN. Instead of modifying the DNNs's architecture, other approaches [18, 37] incorporate an auxiliary "out-of-distribution" (OoD) dataset during training, which is disjoint from the actual training dataset. These ideas have been employed for anomaly segmentation in [11, 13, 38].

A recent line of work performs anomaly segmentation via generative models that reconstruct / resynthesize the original input image. The intuition is that the reconstructed images will better preserve the appearance of regions containing known objects than those with unknown ones. Pixel-wise anomaly detection is then performed by identifying the discrepancies between the original and reconstructed image. This approach has been used not only for anomaly segmentation [14, 39, 40] but also specifically for road obstacle detection [41, 42, 43].

It is important to note that there are some related works with different definitions of anomaly segmentation. For example, [44] evaluates the segmentation of industrial production anomalies like scratches, and in medical contexts anomaly segmentation can be understood as the detection of diseased parts on e.g. tomography images [45] or brain MRIs [46]. What we define as anomaly segmentation will be discussed in detail in the next Section 3.

## 3 Benchmark Description

The aim of our benchmark is two-fold. On one hand, by providing diverse data with consistent annotations, we seek to facilitate progress in general semantic anomaly segmentation research. On the other hand, by focusing on road scenes, we expect our benchmark to accelerate the progress towards much needed segmentation/obstacle-detection methods for safe automated driving.

To achieve these goals, our benchmark covers two tasks. First, it tackles the general problem of anomaly segmentation, aiming to identify the image regions containing object classes that have never been seen during training, and thus for which semantic segmentation cannot be correct. This is necessary for any reliable decision making process and it is of great importance to many computer vision applications. Note that, in accordance to [7, 8], we define anomaly as objects that do not fit any of the class definitions in the training data. In some works, anomaly may be used to describe visually different inputs like *e.g.* a car in a novel color, which does not fit our definition.

This strict definition of semantic anomalies, however, can sometimes be ill-defined because (i) existing semantic segmentation datasets, such as Cityscapes [1], often contain ambiguous and ignored regions (annotated as *void*), which are not strictly anomalies since they are seen during training; (ii) the boundary of some classes is fuzzy, *e.g.*, cars *vs*. vans *vs*. rickshaws, making it unclear whether some regions should be considered as anomalous or not. To address these issues, and to account for the fact that automated driving systems need to make sure that the road ahead is free of *any* hazardous objects, we further incorporate obstacle segmentation as a second task in our benchmark, whose goal is to identify any non-drivable region on the road, may the non-drivable region correspond to a known object class or an unknown one.

## 3.1 Benchmark Tracks and Datasets

We now present the two tracks in our benchmark, corresponding to the two tasks discussed above. Each track contains its own dataset with different properties and is therefore evaluated separately in our benchmark suite. An overview comparing our datasets to related public ones is given in Table 1.

**RoadAnomaly21.** The road anomaly track benchmarks general anomaly segmentation in full street scenes. It consists of an evaluation dataset of 100 images with pixel-level annotations. The data is an extension of the one introduced in [14], now including a broader collection of images and finer-grain labeling. In particular, we removed low quality images and ones lacking clear road scenes. Besides, we removed labeling mistakes, added the void class and included 68 newly collected images. Each image contains at least one anomalous object, *e.g.*, an animal or an unknown vehicle. The anomalies can appear anywhere in the image, which were collected from web resources and therefore depict a wide variety of environments. The distribution of object sizes and location is shown in Figure 2(a). Moreover, we provide 10 additional images with annotations such that users can check the compatibility of their methods with our benchmark implementation.

**RoadObstacle21.** The road obstacle track focuses on safety for automated driving. The objects to segment in the evaluation data always appear on the road ahead, *i.e.* they represent realistic and hazardous obstacles that are critical to detect. Our dataset consists of 222 new images taken by ourselves and 105 from [42], summing up to a total of 327 evaluation images with pixel-level annotations. The region of interest in these images is given by the road, which is assumed to belong to the known classes on which the algorithm was trained. The obstacles in this dataset are chosen such that they all can be understood as anomalous objects as well, *e.g.*, stuffed toys, sleighs or tree stumps. They appear at different distances (one distance per image) and are surrounded by road pixels. This allows us to focus our evaluation on the obstacles, as other objects lie outside the region of interest. The distribution of object sizes and location is shown in Figure 2(b). Moreover, this dataset incorporates different road surfaces, lighting and weather conditions, thus encompassing a broad diversity of scenes. An extra track of additional 85 images with scenes at night and in extreme weather, such as snowstorms, is also available. However, the latter subset is excluded from our numerical experiments due to the significant domain shift. Lastly, we provide 30 additional images with annotations such that users can check the compatibility of their methods with our benchmark implementation.

**Labeling Policy.** In both datasets, the pixel-level annotations include three classes: **1)** anomaly / obstacle, **2)** not anomaly / not obstacle, and **3)** void.

In RoadAnomaly21, the 19 Cityscapes evaluation classes [1], on which most semantic segmentation DNNs are trained, serve as basis to judge whether an object is considered anomalous or not. Everything that fits in the class definitions of Cityscapes is thus labeled as *not anomaly*. This track focuses on the detection of objects which are semantically different from those in the Cityscapes training data. Therefore, if image regions cannot be clearly assigned to any of the Cityscapes classes, they are labeled as *anomaly*. The objects, which are not the main anomalies of interest in the context of street scenes, are labeled as *void* and excluded from our evaluation. The latter class include, for instance, mountains or water in the image background, and street lights. In ambiguous cases, which *e.g.* can arise from a strong domain shift to Cityscapes, we assign the void class as well to properly evaluate semantic anomaly segmentation.

In RoadObstacle21, the task is defined as distinguishing between drivable area and non-drivable area. The goal is to make sure that the road ahead of the ego-car is free of any hazard, irrespective of the object category of potential obstacles. Therefore, the drivable area is labeled as *not obstacle*. This class particularly also includes regions on the road, which visually differ from the rest of the road.

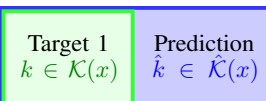 IoU$(k) = 0.50$
sIoU$(k) = 0.50$ 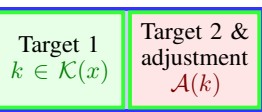 IoU$(k) = 0.50$
sIoU$(k) = 0.99$

Figure 3: Illustration of the ordinary component-wise intersection over union (IoU) and the adjusted one (sIoU). In both examples above, the prediction $\hat{k}$ (blue rectangle) is the same but covers different targets (green rectangles). On the left, both IoU and sIoU yield the same score. On the right, IoU punishes the prediction as it does not cover each object precisely. By contrast, sIoU checks how much the predictions cover the ground-truth regions, independently of whether prediction/ground truth belongs to a single or multiple objects. In automated driving, it is more important to detect all anomalous regions (whether they belong to single or multiple objects), rather than to detect each object precisely. Since two targets are separated by at least one pixel, IoU = sIoU = 1 if and only if the prediction covers one target perfectly.

Moreover, every object, which is visually enclosed in the drivable area, is labeled as *obstacle*. All image regions outside the road are assigned to the *void* class and ignored in the evaluation.

As a quality assessment for both tracks, each labeled image was reviewed by at least three people in order to guarantee the highest quality of labels.

## 3.2 Performance Metrics

For the sake of brevity, in what follows we refer to both anomalies and obstacles as *anomalies*.

**Pixel level.** Let $\mathcal{Z}$ denote the set of image pixel locations. A model with a binary classifier providing anomaly scores $s(x) \in \mathbb{R}^{|\mathcal{Z}|}$ for an image $x \in \mathcal{X}$ (from a dataset $\mathcal{X} \subseteq [0,1]^{N \times |\mathcal{Z}| \times 3}$ of $N$ images) discriminates between the two classes *anomaly* and *non-anomaly*. We evaluate the separability of the pixel-wise anomaly scores via the area under the precision-recall curve (AuPRC), where precision and recall are considered as functions of some threshold $\delta \in \mathbb{R}$ applied to $s(x) \ \forall \ x \in \mathcal{X}$. The AuPRC puts emphasis on detecting the minority class, making it particularly well suited as our main pixel-wise evaluation metric since the pixel-wise class distributions of RoadAnomaly21 and RoadObstacle21 are considerably unbalanced, *c.f*. Table 1.

To consider the safety point of view, we also include the false positive rate at 95% true positive rate (FPR$_{95}$) in our evaluation. The FPR$_{95}$ metric indicates how many false positive predictions must be made to reach the desired true positive rate. Note that, any prediction which is contained in a ground-truth labeled region of the class void is not counted as false positive, *c.f*. Section 3.1. In particular for the RoadObstacle21 dataset the evaluation is therefore restricted to the road area.

**Component level.** From a practitioner's perspective, it is very important to detect all anomalous regions in the scene, regardless of their size, *i.e*., the number of pixels they cover. However, pixel-level metrics may neglect small anomalies. While one could thus focus on object detection metrics, the notion of individual objects is in fact not relevant for anomaly (region) detection. To satisfy these requirements, we therefore consider performance metrics acting at the component level.

The main metrics for component-wise evaluation are the numbers of *true-positives* (TP), *false-negatives* (FN) and *false-positives* (FP). Considering anomalies as the positive class, we use a component-wise localization and classification quality measure to define the TP, FN and FP components. Specifically, we define this measure as an adjusted version of the component-wise intersection over union (sIoU), introduced in [47]. In particular, while in [47] the sIoU is computed for predicted components , we consider the sIoU for ground-truth components to compute TP and FN. To compute FP, we employ the positive predictive value (PPV, or component-wise precision) for predicted components as quality measure. We discuss the definitions of these quantities in more detail below.

Let $\mathcal{Z}_c$ be the set of pixel locations labeled with class $c =$ "anomaly" in the dataset $\mathcal{X}$. We consider a connected component of pixels (where the 8 pixels surrounding pixel $z$ in image $x \in \mathcal{X}$ are taken to be its neighbors) that share the same class label as a *component*. Then, let us denote by $\mathcal{K} \subseteq \mathcal{P}(\mathcal{Z}_c)$, with $\mathcal{P}(\mathcal{S})$ the power set of a set $\mathcal{S}$, the set of anomaly components according to the ground truth, and by $\hat{\mathcal{K}} \subseteq \mathcal{P}(\mathcal{Z}_c)$ the set of components predicted to be anomalous by some machine learning model.

Formally, the sIoU is a mapping $\text{sIoU} : \mathcal{K} \to [0, 1]$. For $k \in \mathcal{K}$, it is defined as

$$\text{sIoU}(k) := \frac{|k \cap \hat{K}(k)|}{|(k \cup \hat{K}(k)) \setminus \mathcal{A}(k)|} \quad \text{with} \quad \hat{K}(k) = \bigcup_{\hat{k} \in \hat{\mathcal{K}}, \hat{k} \cap k \neq \emptyset} \hat{k} \tag{1}$$

and $\mathcal{A}(k) = \{z \in k' : k' \in \mathcal{K} \setminus \{k\}\}$. With the adjustment $\mathcal{A}(k)$, the pixels are excluded from the union if and only if they correctly intersect with another ground-truth component $k' \in \mathcal{K}(x)$, which is not equal to $k$. This may happen when one predicted component covers multiple ground-truth components, as illustrated in Figure 3. Given some threshold $\tau \in [0, 1)$, we then call a target $k \in \mathcal{K}$ TP if $\text{sIoU}(k) > \tau$, and FN otherwise. We refer to Appendix C.2 for qualitative examples of the difference between IoU and sIoU.

For the other error type, *i.e.*, FP, we compute the PPV (or precision) for $\hat{k} \in \hat{\mathcal{K}}$, which is defined as

$$\text{PPV}(\hat{k}) := \frac{|\hat{k} \cap K(\hat{k})|}{|\hat{k}|} , \tag{2}$$

We then call a predicted component $\hat{k} \in \hat{\mathcal{K}}$ FP if $\text{PPV}(\hat{k}) \leq \tau$.

As an overall metric, we additionally include the component-wise $F_1$-score defined as

$$F_1(\tau) := \frac{2 \cdot \text{TP}(\tau)}{2 \cdot \text{TP}(\tau) + \text{FN}(\tau) + \text{FP}(\tau)} \in [0, 1] , \tag{3}$$

which summarizes the TP, FN and FP quantities (that depend on $\tau$). The component-level metrics allow one to evaluate localization of objects irrespective of their size and hence big objects will not dominate these metrics. In addition, while object detection metrics punish predictions that cover multiple ground-truth objects or vice-versa, our component-level metric does not do so, *c.f.* Figure 3.

### 3.3 Evaluated Methods

Several anomaly segmentation methods have already been evaluated on our benchmark and constitute our initial leader board. We evaluate at least one method per type discussed in Section 2.2, namely

- *Methods originating from image classification*: **maximum softmax probability** [23], **ODIN** [22], **Mahalanobis distance** [21];
- *Bayesian model uncertainty*: **Monte Carlo (MC) dropout** [35], **ensemble** [32];
- *Learning to identify anomalies*: **learned embedding density** [7], **void classifier** [7], **maximized softmax entropy** [11];
- *Reconstruction via generative models*: **image resynthesis** [14], **SynBoost** [39] and **road inpainting** (obstacle track only) [42].

All methods have an underlying semantic segmentation DNN trained on Cityscapes and provide pixel-wise anomaly scores. A semantic segmentation DNN trained on Cityscapes is also our recommendation as underlying model, however, we leave it up to the participants which network and training data they use. Furthermore, some evaluated methods additionally employ out-of-distribution (OoD) data to tune the anomaly detector. For our set of methods, this would be any data with labels semantically different from the Cityscapes train classes. OoD data is also allowed to be used to alleviate the effects of a potential domain shift. For additional details on the methods, we refer the reader to Appendix D.

## 4 Numerical Experiments

In our benchmark suite we integrate a default method to generate the anomaly segmentation from pixel-wise anomaly scores. We choose the threshold $\delta^*$, at which one pixel is classified as anomaly, by means of the optimal pixel-wise $F_1$-score, that we denote with $F_1^*$. Then, $\delta^*$ is computed as

$$\delta^* = \arg\max_{\delta \in \mathbb{R}} \; 2 \cdot \text{precision}(\delta) \cdot \text{recall}(\delta) \; / \; (\text{precision}(\delta) + \text{recall}(\delta)) , \tag{4}$$

subject to $\text{precision}(\delta) + \text{recall}(\delta) > 0 \; \forall \; \delta$. In Appendix E we provide a study where $\delta^*$ is varied.

| | | Pixel-level | | | Component-level | | | | | | | | | |
|---|---|---|---|---|---|---|---|---|---|---|---|---|---|---|
| | | Anomaly scores | | | $k \in \mathcal{K}$ | $\hat{k} \in \hat{\mathcal{K}}$ | $\tau = 0.25$ | | | $\tau = 0.50$ | | | $\tau = 0.75$ | | |
| Method | requires OoD data | AuPRC ↑ | FPR$_{95}$ ↓ | $F_1^*$ ↑ | $\overline{\text{sIoU}}$ ↑ | $\overline{\text{PPV}}$ ↑ | FN ↓ | FP ↓ | $F_1$ ↑ | FN ↓ | FP ↓ | $F_1$ ↑ | FN ↓ | FP ↓ | $F_1$ ↑ | $\overline{F_1}$ ↑ |
| Maximum softmax [23] | ✗ | 28.0 | 72.0 | 34.2 | 15.5 | 15.3 | 204 | 681 | 11.6 | 233 | 714 | 5.8 | 256 | 744 | 1.2 | 5.9 |
| ODIN [22] | ✗ | 33.1 | 71.7 | 39.1 | 19.6 | 17.9 | 181 | 924 | 12.8 | 226 | 985 | 5.6 | 254 | 1043 | 1.2 | 6.0 |
| Mahalanobis [21] | ✗ | 20.0 | 87.0 | 31.9 | 14.8 | 10.2 | 206 | 1433 | 6.4 | 241 | 1478 | 2.4 | 257 | 1512 | 0.6 | 2.9 |
| MC dropout [35] | ✗ | 28.9 | 69.5 | 39.0 | 20.5 | 17.3 | 175 | 1320 | 10.4 | 225 | 1391 | 4.4 | 252 | 1459 | 1.2 | 4.9 |
| Ensemble [32] | ✗ | 17.7 | 91.1 | 27.8 | 16.4 | 20.8 | 197 | 1454 | 7.3 | 233 | 1511 | 3.2 | 254 | 1553 | 0.9 | 3.4 |
| Void classifier [7] | ✓ | 36.8 | 63.5 | 44.3 | 21.1 | 22.1 | 181 | 797 | 14.2 | 219 | 845 | 7.5 | 253 | 879 | 1.6 | 7.6 |
| Embedding density [7] | ✗ | 37.5 | 70.8 | 48.7 | 33.8 | 20.5 | 107 | 1437 | 16.7 | 176 | 1485 | 9.4 | 250 | 1592 | 1.3 | 9.2 |
| Image resynthesis [14] | ✗ | 52.3 | 25.9 | 60.5 | 39.5 | 11.0 | 95 | 1187 | 20.7 | 153 | 1225 | 13.7 | 230 | 1294 | 4.0 | 12.9 |
| SynBoost [39] | ✓ | 56.4 | 61.9 | 58.0 | 35.0 | 18.3 | 109 | 1062 | 20.7 | 178 | 1114 | 11.5 | 247 | 1216 | 2.0 | 11.5 |
| Maximized entropy [11] | ✓ | **85.5** | **15.0** | **77.4** | **49.2** | **39.5** | **85** | **413** | **41.5** | **115** | **421** | **35.4** | **163** | **439** | **24.8** | **34.5** |

Table 2: Benchmark results for our RoadAnomaly21 dataset. This dataset contains 262 ground-truth components in total. The main performance metrics are highlighted with gray columns.

| | | Pixel-level | | | Component-level | | | | | | | | | |
|---|---|---|---|---|---|---|---|---|---|---|---|---|---|---|
| | | Anomaly (obstacle) scores | | | $k \in \mathcal{K}$ | $\hat{k} \in \hat{\mathcal{K}}$ | $\tau = 0.25$ | | | $\tau = 0.50$ | | | $\tau = 0.75$ | | |
| Method | requires OoD data | AuPRC ↑ | FPR$_{95}$ ↓ | $F_1^*$ ↑ | $\overline{\text{sIoU}}$ ↑ | $\overline{\text{PPV}}$ ↑ | FN ↓ | FP ↓ | $F_1$ ↑ | FN ↓ | FP ↓ | $F_1$ ↑ | FN ↓ | FP ↓ | $F_1$ ↑ | $\overline{F_1}$ ↑ |
| Maximum softmax [23] | ✗ | 15.7 | 16.6 | 22.5 | 19.7 | 15.9 | 255 | 1494 | 13.2 | 326 | 1503 | 6.3 | 372 | 1517 | 1.7 | 6.9 |
| ODIN [22] | ✗ | 21.2 | 15.4 | 29.2 | 20.7 | 18.5 | 260 | 1072 | 16.1 | 312 | 1079 | 9.9 | 362 | 1093 | 3.5 | 10.0 |
| Mahalanobis [21] | ✗ | 20.9 | 13.1 | 25.8 | 14.0 | 21.8 | 293 | 1101 | 12.0 | 352 | 1104 | 4.7 | 385 | 1116 | 0.4 | 5.5 |
| MC dropout [35] | ✗ | 3.7 | 50.6 | 8.0 | 6.3 | 5.8 | 351 | 2782 | 2.3 | 375 | 2784 | 0.8 | 386 | 2790 | 0.1 | 1.0 |
| Ensemble [32] | ✗ | 1.1 | 77.2 | 3.1 | 8.6 | 4.7 | 335 | 3758 | 2.5 | 365 | 3768 | 1.1 | 382 | 3782 | 0.3 | 1.3 |
| Void classifier [7] | ✓ | 9.2 | 41.5 | 23.4 | 6.3 | 20.3 | 350 | 350 | 9.8 | 365 | 350 | 6.0 | 381 | 353 | 1.9 | 5.9 |
| Embedding density [7] | ✗ | 0.8 | 46.4 | 2.0 | 35.6 | 2.9 | 145 | 10972 | 4.2 | 244 | 11037 | 2.5 | 370 | 11191 | 0.3 | 2.4 |
| Image resynthesis [14] | ✗ | 37.2 | 4.7 | 38.8 | 16.6 | 20.5 | 286 | 743 | 16.5 | 334 | 773 | 8.9 | 374 | 824 | 2.3 | 9.5 |
| Road inpainting [42] | ✗ | 52.6 | 47.1 | 67.5 | **57.6** | 39.5 | **79** | 580 | 48.4 | **131** | 586 | 41.8 | **240** | 611 | 25.8 | 40.2 |
| SynBoost [39] | ✓ | 70.3 | 3.1 | 70.1 | 44.3 | 41.8 | 133 | 352 | 51.3 | 185 | 363 | 42.6 | 286 | 414 | 22.6 | 40.4 |
| Maximized entropy [11] | ✓ | **85.1** | **0.8** | **79.6** | 47.9 | **62.6** | 136 | **151** | **63.7** | 177 | **158** | **55.7** | 247 | **174** | **40.1** | **54.2** |

Table 3: Benchmark results for our RoadObstacle21 dataset. This dataset contains 388 ground-truth components in total. The main performance metrics are highlighted with gray columns.

Moreover, for the anomaly track, components smaller than 500 pixels are discarded from the predicted segmentation, and for the obstacle track, components smaller than 50 pixels are discarded. These sizes are chosen based on the smallest ground-truth components. All methods presented in Section 3.3 produce anomaly scores for which we apply the default segmentation method. We emphasize that using our proposed default method for anomaly segmentation masks is completely optional. We provide results without filtering by predicted component sizes in Appendix E. We allow and encourage competitors in the benchmark to submit their own anomaly segmentation masks generated via more sophisticated image operations.

In our evaluation, we additionally include the average sIoU per component $\overline{\text{sIoU}}$, which can be computed by averaging sIoU over all ground-truth components $k \in \mathcal{K}$. Analogously, we also include the average PPV per component $\overline{\text{PPV}}$ for all predicted components $\hat{k} \in \hat{\mathcal{K}}$. As the number of component-wise TP, FN and FP depends on some threshold $\tau$ for sIoU and PPV, respectively (see Section 3.2), we average these quantities over different thresholds $\tau \in \mathcal{T} = \{0.25, 0.30, \ldots, 0.75\}$, similarly to [4], yielding the averaged component-wise $F_1$ score $\overline{F_1} = \frac{1}{|\mathcal{T}|} \sum_{\tau \in \mathcal{T}} F_1(\tau)$.

**Discussion of the Results.** Our benchmark results for RoadAnomaly21 and RoadObstacle21 are summarized in Table 2 and Table 3, respectively. In general, we observe that methods originally designed for image classification, including maximum softmax, ODIN, and Mahalanobis, do not generalize well to anomaly and obstacle segmentation. For methods based on statistics of the Cityscapes dataset, such as Mahalanobis as well as learned embedding density, anomaly detection is typically degraded by the presence of a domain shift. This results in a poor performance, particularly in RoadObstacle21, where various road surfaces can be observed. Interestingly, learned embedding density, MC dropout and the void classifier yield worse performance than maximum softmax on RoadObstacle21, whereas we observe the opposite on RoadAnomaly21.

The detection methods based on generative models, namely image resynthesis and SynBoost, appear to be better suited to both anomaly and obstacle segmentation at pixel as well as component level, clearly being superior to all the approaches discussed previously. This observation also holds for road inpainting in the obstacle track. These autoencoder-based methods are nonetheless limited by their

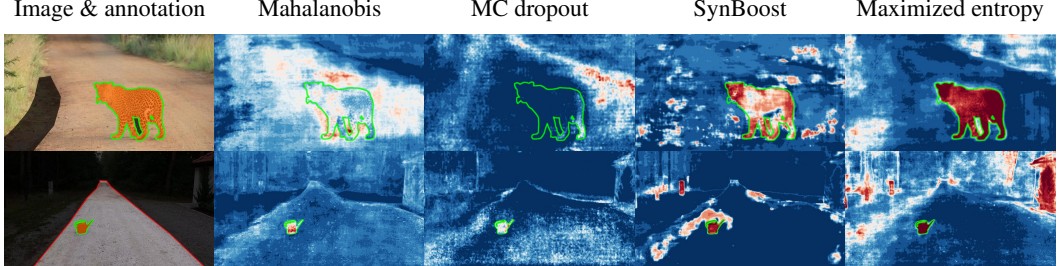

| Image & annotation | Mahalanobis | MC dropout | SynBoost | Maximized entropy |

Figure 4: Qualitative comparison of the anomaly scores produced by the methods introduced in Section 3.3 for one example image of RoadAnomaly21 (top row) and one example image of RoadObstacle21 (bottom row). Here, red indicates higher anomaly / obstacle scores and blue lower ones. The ground-truth anomaly / obstacle component is indicated by green contours.

|  |  | RoadAnomaly | | Fishyscapes LostAndFound Validation | | | | | RoadObstacle | | LostAndFound test-NoKnown | | | | |
|  |  |  |  | Pixel-level | | Component-level | | |  |  | Pixel-level | | Component-level | | |
|  |  |  |  | Anomaly scores | | $k \in \mathcal{K}$ | $\hat{k} \in \hat{\mathcal{K}}$ |  |  |  | Anomaly scores | | $k \in \mathcal{K}$ | $\hat{k} \in \hat{\mathcal{K}}$ |  |
| Method | OoD data | AuPRC↑ | $\overline{F_1}$↑ | AuPRC↑ | FPR$_{95}$↓ | sIoU↑ | $\overline{PPV}$↑ | $\overline{F_1}$↑ | AuPRC↑ | $\overline{F_1}$↑ | AuPRC↑ | FPR$_{95}$↓ | sIoU↑ | $\overline{PPV}$↑ | $\overline{F_1}$↑ |
| Maximum softmax [23] | ✗ | 28.0 | 5.9 | 5.6 | 40.5 | 3.5 | 9.5 | 1.8 | 15.7 | 6.9 | 30.1 | 33.2 | 14.2 | 62.2 | 13.4 |
| ODIN [22] | ✗ | 33.1 | 6.0 | 15.5 | 38.4 | 9.9 | 21.9 | 9.7 | 21.2 | 10.0 | 51.0 | 30.7 | 38.9 | 48.0 | 38.1 |
| Mahalanobis [21] | ✗ | 20.0 | 2.9 | 32.9 | **8.7** | 19.6 | 29.4 | 19.2 | 20.9 | 5.5 | 55.0 | 12.9 | 33.8 | 31.7 | 24.6 |
| MC dropout [35] | ✗ | 28.9 | 4.9 | 14.4 | 47.8 | 4.8 | 18.1 | 4.3 | 3.7 | 1.0 | 36.2 | 36.0 | 17.0 | 34.7 | 14.7 |
| Ensemble [32] | ✗ | 17.7 | 3.4 | 0.3 | 90.4 | 3.1 | 1.1 | 0.4 | 1.1 | 1.3 | 2.9 | 82.0 | 6.7 | 7.6 | 2.7 |
| Void classifier [7] | ✓ | 36.8 | 7.6 | 11.7 | 15.3 | 9.2 | 39.1 | 14.9 | 9.2 | 5.9 | 4.4 | 47.0 | 0.7 | 35.1 | 1.1 |
| Embedding density [7] | ✗ | 37.5 | 9.2 | 8.9 | 42.2 | 5.9 | 10.8 | 4.9 | 0.8 | 2.4 | 61.7 | 10.4 | 37.8 | 35.2 | 30.8 |
| Image resynthesis [14] | ✗ | 52.3 | 12.9 | 5.1 | 29.8 | 5.1 | 12.6 | 4.1 | 37.2 | 9.5 | 57.1 | 8.8 | 27.2 | 30.7 | 21.5 |
| Road inpainting [42] | ✗ | - | - | - | - | - | - | - | 52.6 | 40.2 | **83.0** | 35.7 | **49.2** | 60.7 | **56.9** |
| SynBoost [39] | ✓ | 56.4 | 11.5 | **64.9** | 30.9 | **27.9** | 48.6 | **38.0** | 70.3 | 40.4 | 81.8 | **4.6** | 37.2 | **72.3** | 53.0 |
| Maximized entropy [11] | ✓ | **85.5** | **34.5** | 44.3 | 37.7 | 21.1 | 48.6 | 30.0 | **85.1** | **54.2** | 77.9 | 9.7 | 45.9 | 63.1 | 55.0 |

Table 4: Benchmark results for Fishyscapes LostAndFound validation and LostAndFound test-NoKnown, containing 165 and 1709 ground-truth components in total, respectively. In this table the pixel-wise AuPRC and the component-wise $\overline{F_1}$ from RoadAnomaly21 and RoadObstacle21, *c.f.* Table 2 and Table 3, are additionally included for cross evaluation (gray columns).

discrepancy module, and they are outperformed in our experiments by maximized softmax entropy, which peaks at an AuPRC of 86% and a component-wise $\overline{F_1}$ of 49%. This highlights the importance of anomaly and obstacle proxy data. Illustrative example score maps produced by the discussed methods are shown in Figure 4.

In summary, the component-level evaluation highlights the methods' weaknesses even more clearly than the pixel-wise evaluation, the latter giving a stronger weight to larger anomalies and obstacles. All methods indeed tend to face difficulties in the presence of smaller anomalies and obstacles, as we demonstrate in more detail in Appendix H. In addition, we observe a much lower component-wise $\overline{F_1}$ score than a pixel-wise $F_1^*$, demonstrating the importance of evaluating at component level. The results w.r.t. the different categories of methods are challenging for models, hence leaving room for improvement.

Our benchmark suite enables a unified evaluation across different datasets whenever ground truth is available. In Table 4 we summarize our results for Fishyscapes LostAndFound [7], a validation set of 100 LostAndFound images [25] with refined labels fitting the anomaly track, and the LostAndFound test split, with original labels fitting the obstacle track. Note that, for the LostAndFound test split, we filtered out all images that contain humans and bicycles labeled as obstacles (therefore called LostAndFound test-NoKnown) because we applied anomaly segmentation methods out of the box to the task of obstacle segmentation, and these methods focus on previously-unseen objects.

In comparison to our datasets, for both LostAndFound datasets we observe a less pronounced gap, in terms of both main performance metrics, the pixel-level AuPRC and component-level $\overline{F_1}$ scores, between the methods orignally designed for image classification, especially ODIN and Mahalanobis, and those specifically designed for anomaly segmentation, especially road inpainting and maximized entropy. This signals that both of our datasets contribute new challenges for anomaly and obstacle

segmentation. In Appendix H and Appendix I we provide further and more fragmented results in terms of both objects sizes and object categories.

Finally, we also applied our benchmark suite to the LiDAR guided Small obstacle Segmentation dataset [26]. Our main findings are that our whole set of methods yields weak performance on that dataset. The main purpose of this dataset is the detection of small obstacles from multiple sensors including LiDAR. Hence, the conditions for the other sensor modalities are purposely challenging (*e.g.*, low illumination), making this dataset less suitable to camera-only methods. We present the corresponding results in Appendix F.4.

## 5 Conclusion

In this work, we have introduced a unified and publicly available benchmark suite that evaluates a method's performance for anomaly segmentation with established pixel level as well as recent component level metrics. Our benchmark suite is applicable in a plug and play fashion to any dataset for anomaly segmentation that comes with ground truth, such as LostAndFound and Fishyscapes LostAndFound, allowing for a better comparison of new methods. Moreover, our benchmark is accompanied with two publicly available datasets, RoadAnomaly21 for anomaly segmentation and RoadObstacle21 for obstacle segmentation.

These two datasets challenge two important abilities of computer vision systems: On one hand their ability to detect and localize unknown objects; on the other hand their ability to reliably detect and localize obstacles on the road, may they be known or unknown. Our datasets consist of real images with pixel-level annotations and depict street scenes with higher variability in object types and object sizes than existing datasets. Our experiments have demonstrated that both of our datasets show a distinct separation in terms of performance between the methods that are specifically designed for anomaly / obstacle segmentation and those that are not. However, there remains much room for performance improvement, particularly in terms of component-wise metrics, which stresses the need for future research in the direction of anomaly segmentation.

The images of the datasets and the software are available at https://www.segmentmeifyoucan.com/.

## Broader Impact

This benchmark advances research towards the safe deployment of autonomous vehicles. This ultimately will have many consequences, *e.g.*, reducing the number of jobs in the transport sector. More immediately, the benchmark measures the reliability of algorithms and therefore may be misunderstood as giving safety guarantees. This benchmark however only works for the specified training regime *i.e.* it cannot certify fitness for real-world deployment and should not be misunderstood as such. In particular, while our datasets greatly contribute to the diversity of anomalies, the scale of the datasets is still not even close to sufficient in order to represent every possible type of an anomaly. Furthermore, although we do not publicly provide test labels, there remains a risk, common to any other benchmark, of the community designing methods that overfit on our benchmark tasks.

## Acknowledgement

Robin Chan and Svenja Uhlemeyer acknowledge funding by the German Federal Ministry for Economic Affairs and Energy, within the projects "KI Absicherung - Safe AI for Automated Driving", grant no. 19A19005R, and "KI Delta Learning - Scalable AI for Automated Driving", grant no. 19A19013Q, respectively. We thank the consortiums for the successful cooperation. We would also like to thank the "BUW-KI" team who substantially contributed to collecting and labeling of data.

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
