# OpenReview forum: "SegmentMeIfYouCan: A Benchmark for Anomaly Segmentation"
_NeurIPS.cc/2021/Track/Datasets_and_Benchmarks/Round2 — NeurIPS 2021 Datasets and Benchmarks Track (Round 2)_

### Official Review · Reviewer_XLs8 · 2021-09-13
**An interesting dataset for road scene anomaly segmentation, but there are some issues**

**Rating:** 6
**Confidence:** 4
**Clarity:** The paper is well-written and easy to…

**Strengths:**

The proposed benchmark addresses an important task, for which there are not a lot of datasets available. The images in the datasets are very diverse. Each image shows a different scene. (This is in contrast to some other datasets for anomaly segmentation in road scenes, which include images that are almost the same since they are frames taken from a video of a single scene). RoadAnomaly21 and RoadObstacle21 also have a large variety with respect to object types, sizes, and distances.

The differentiation between the pure anomaly segmentation task and the obstacle segmentation task appears to not have been done before.

The experiments and evaluation are very thorough.

**Weaknesses:**

The benchmark only contains images for evaluation, making it necessary to train a model on other datasets (usually CityScapes). This leads to a potential domain-shift that complicates the interpretation of results. If a model performs badly on the benchmark (in particular if it predicts a lot of false positives), it is unclear whether the bad performance is a flaw of the anomaly segmentation method or caused by the domain shift. __Edited to add:__ The authors clarified their definition of "anomaly" and emphasized that a certain amount of domain shift is intentional. They want a method to segment only pixels that belong to semantically new classes. Pixels with visually new features that belong to one of the training classes should not be segmented as anomalies. I think this is a valid justification for the present domain shift. On the other hand, the domain shift still makes it difficult to analyse the reason for false positives of a method: It would be unclear whether the false positives are caused by the domain shift or by issues of the method itself.

**Additional Feedback:**

In addition to the results on the anomaly segmentation benchmark, it might be a good idea to also report the results of the methods on CityScapes. That way, a reader would get a more complete picture.

I think there is a typo in equation (1). I think it should be $sIoU(k) := \frac{|k\cap\hat K(k)|}{|(k\cup \hat K(k))\backslash\mathcal{A}(k)|}$.

The "Checklist" section at the end of the paper begins with an instructions block. That block can be removed.

**Correctness:**

On the whole, the dataset construction and evaluation protocol are appropriate. I have two issues, however:

I find the use of the void class problematic. While I understand the motivation (e.g., mountains in the background are not of interest for the task in question), it uses information that would not be available in a real application. The voided areas may contain false positive or negative predictions of the model that could affect the application. I found a particularly illustrative example of my issues with the void class in the presentation (https://segmentmeifyoucan.github.io/segment-me-presentation/#/10) linked on the official benchmark website (https://segmentmeifyoucan.com/): There it is noted that wet spots on the road are ignored in the evaluation. However, the predictions of a model in these spots are very much of interest, as the model might falsely predict obstacles in these regions. Also from a more theoretical standpoint, when using the dataset to evaluate an anomaly segmentation method, a user would want to test whether a model correctly detects all kinds of anomalies, not just a somewhat arbitrary subset. __Edited to add__: The authors addressed this issue in their reply. They explained that they revised the annotations for RoadObstacle21 so that parts of the road that may contain visually new features, but are still drivable (such as wet spots), are marked as obstacle-free. From an application point of view, I still find the void class to be problematic (e.g., semantically ambiguous objects such as rickshaws cannot be ignored in real applications), but I acknowledge that the limitations of the CityScapes annotations and classes cause this problem.

At the beginning of Section 4, "Numerical Experiments", the authors write that for RoadAnomaly21, components smaller than 500 pixels are discarded from the predicted segmentation (50 pixels for RoadObstacle21). This choice is somewhat questionable. It penalizes methods that produce predictions that cover ground truth components well, but consist of a lot of small components. Conversely, it does not penalize methods that predict lots of small false positive regions (that might potentially cover a significant area of the image). __Edited to add:__ In their reply, the authors explain that the removal of small predicted connected components is not part of the evaluation protocol, but an optional choice made for the methods. I think it is legitimate to include such a postprocessing step in a method, so my original issue is sufficiently addressed. There is, however, still a problem, as the choice to remove connected components smaller than 500/50 pixels is based on knowledge of the size of ground truth connected components.

**Documentation:**

The datasets are described in detail. They are hosted on multiple mirrors. The given benchmark website contains download links, a leaderboard, a link to the evaluation code and other information. The supplementary material contains a dataset sheet with detailed information about the dataset.

From a cursory glance, it seems as if the provided code should make it possible to repoduce the results.

**Ethics:**

No additional ethics review appears to be necessary.

**Relation To Prior Work:**

The differences of the proposed datasets to other anomaly segmentation datasets in the domain of road scenes are discussed in detail. However, there is no mention of anomaly segmentation datasets in other domains.

In the paper, the authors use the term anomaly segmentation in the context of models that have been trained for semantic segmentation (with a plurality of classes) that should also segment new anomalous classes. However, the term anomaly segmentation is also used for problems where models are trained on data that is not labelled as containing several classes. The "Related Work" section could benefit from a discussion that addresses this difference. This distinction is not just important for the datasets, but also for methods. While I assume that the authors designed their benchmark for the evaluation of methods that have been trained to segment different classes, this is not explicitly stated. If my assumption is correct, this should be made clear in the paper. Otherwise, the "Related Work" section (and the experiments) should also include recent methods that train only on "non-anomalous" data without class distinction. __Edited to add:__ The authors sufficiently addressed this point, as they clarified their definition of "anomaly" in the paper.

The discussion of anomaly segmentation methods that are trained for semantic segmentation is detailed. While ensemble methods are cited in the "Related Work" section, they were not included in the evaluation. This is a bit surprising, since ensemble methods are consistently among the best methods for anomaly detection in a classification setting. __Edited to add:__ I their reply, the authors addressed this point sufficiently. They explain that a problem with ensembles is the associated computational complexity, which is why they are not an established method for semantic segmentation. Additionally, the authors stated that they will perform additional experiments with ensembles so that they can be included in the paper.

**Summary And Contributions:**

The paper introduces the "SegmentMeIfYouCan" benchmark that contains two datasets for anomaly and obstacle segmentation in road scenes. The first dataset, RoadAnomaly21, can be used to evaluate the capability of semantic segmentation models to segment anomalous objects, i.e., objects not seen during training. The second dataset, RoadObstacle21, is aimed at obstacle segmentation, where a model is supposed to segment all objects on the road, regardless of whether they have been seen before or not.

RoadAnomaly21 contains 100 images of road scenes with pixel-level annotation for three classes: All pixels that are part of one of the 19 CityScapes evaluation classes are labelled as "not anomalous". Pixels in objects that do not belong to a CityScapes class are labelled as "anomalous". Certain parts of the images that the authors do not consider to be main anomalies of interest in the context of street scenes, such as mountains in the image background and street lights, are labelled as "void", meaning they will be ignored in the evaluation.

RoadObstacle21 contains 327 images showing road scenes with a variety of obstacles on the road (e.g., stuffed toys, tree stumps). These objects appear at different distances in the images. In the images, the drivable area is labeled as "not obstacle". All objects that are enclosed by the drivable area are labelled as "obstacle". Pixels outside of the road are labelled as "void" and therefore ignored in the evaluation. RoadObstacle21 also contains 85 images with scenes at night and in extreme wheather. These images are not included in the experiments because of the large domain shift.

For both datasets, the labels are withheld by the authors, to avoid the possibility that methods are overfitted to these specific datasets. However, the authors provide small validation sets that include labels, so that users can check the compatibility of their method with the benchmark implementation.

In their benchmark, the authors evaluate a diverse array of methods on the datasets. Some of these use "auxiliary" out-of-distribution data, while others do not. The authors use pixel-level metrics (e.g., AuPRC) as well component-level metrics (a modified version of IoU) in their benchmark. The performance of the evaluated methods differs wildly, with methods based on generative models and the maximized softmax entropy method performing the best. Overall, the evaluation shows that there is significant room for improvement.
The authors also include evaluations of the methods on other datasets, such as Fishyscapes LostAndFound, to complement their benchmark.

---

> ### Author Response · Authors · 2021-09-27
> **Reply to Reviewer XLs8**
>
> First of all thanks for the detailed review.
>
> > I find the use of the void class problematic.... There it is noted that wet spots on the road are ignored in the evaluation. However, the predictions of a model in these spots are very much of interest
>
> We refer to our general comment on domain shift (see top) where we argue why the void-class is necessary in general. The motivation is not to ignore uninteresting areas, but to avoid punishing methods for the limitations of the class definitions. For the example of mountains in the background: These are also labeled as void in Cityscapes. We cannot consider them as anomaly because they are visible in Cityscapes (even when not labeled) and semantically close to other terrain. We can however also not consider them a semantic inlier, because Cityscapes also does not. It would be both acceptable to label mountains as an anomaly, and to not label them as an anomaly. Consequently, we ignore these pixels in our evaluation.
>
> For the RoadObstacle21 dataset, we agree that given the definition of drivable area, wet spots should be considered inliers. As a consequence, we revised the pixel-level annotations of our data. The obstacle track focuses on the task of distinguishing between drivable area (road) and non-drivable area (obstacle). We therefore adjust the labeling policy as follows: we removed the void annotation for areas in the image which are visually different from the road but are still drivable. This is still in line with the motivation of the track, and in particular, this applies to wet spots on the road which are now included as class non-obstacle/drivable in our evaluation. The void class on the drivable area is however not completely removed from the dataset as we still assign that class to regions entirely surrounded by pixels of an obstacle (e.g. the area between a can and its handle is assigned the void class).
>
> ---
> > ...the authors write that for RoadAnomaly21, components smaller than 500 pixels are discarded from the predicted segmentation (50 pixels for RoadObstacle21). This choice is somewhat questionable
>
> All our evaluated methods produce pixel-wise anomaly scores. To evaluate anomaly segmentation, we provide a default method to generate binary segmentation masks. In this context, removing small predicted components, i.e. to use our default method to generate anomaly segmentation masks, is completely optional. The choice is up to the benchmark participants, and in our manuscript we also state that we allow users to submit their own anomaly segmentation masks, which can be generated via more sophisticated operations (line 267-269). However, for our set of methods, we apply the proposed default method as described at the beginning of Section 4 since the method has shown to be advantous in terms of anomaly segmentation accuracy. We will add a table reporting additional results where we evaluate methods without filtering by means of predicted component size.
>
> ---
> > While ensemble methods are cited in the "Related Work" section, they were not included in the evaluation.
>
> Ensembles is a powerful technique in uncertainty quantification for deep learning tasks like image classification or bounding box detection. However, ensembles are not established in semantic segmentation due to the computational complexity. Noticeably, neither Fishyscapes nor CAOS include ensembles in their benchmarks. However, since they have proven to consistently be among best performing methods in classification settings, we will include ensembles as additional baseline. However, the training time of a DeepLabv3+ ensemble might exceed the length of the response period, such that we are unable to update the manuscript in time.
>
> ---
> > it might be a good idea to also report the results of the methods on CityScapes
>
> For anomaly segmentation methods also performing semantic segmentation, we will add a table reporting the mIoU scores on the Cityscapes validation dataset. As summarized in Table 5, we employed three different network architectures in total. The DeepLabv3+ WideResNet38 backbone achieves a mIoU score of 90.3%, DeepLabv3+ Xcpection backbone a mIoU score of 80.3% and PSPNet a mIoU score of 79.9%. We state in line 251 that the choice of underlying semantic segmentation network is up to the participants in the benchmark, any network is allowed to be used.

---

> > ### Comment · Reviewer_XLs8 · 2021-09-29
> > **Reply to the authors**
> >
> > Thank you very much for your comments, which alleviated several of my issues. I edited my review accordingly.

---

### Official Review · Reviewer_j9s3 · 2021-09-17
**Review for the SegmentMeIfYouCan Dataset and Benchmark.**

**Rating:** 5
**Confidence:** 4
**Correctness:** 1. There is a potentially very large …
**Clarity:** The paper is easy to follow and under…

**Strengths:**

1. The paper considers a relevant problem for which existing datasets are not sufficiently large and diverse.
2. The paper is easy to follow and understand.
3. Coverage of related work is adequate. The benchmark contains a diverse set of existing methods and evaluation metrics.

**Weaknesses:**

1. The number of provided images is small. For the anomaly segmentation subset, 100 images are provided. For obstacle detection, 327 images are given. I have concerns that the number of provided images is enough for a thorough method evaluation, especially since the diversity of possible anomalies is immensely large for this particular task.

2. The dataset does not include a training set. The paper also does not contain a clear recommendation on which other datasets should be used for model training. Since objects were defined as anomalous based on the class definitions of the Cityscapes dataset, it seems that using datasets that localize different classes would not be suitable since possible anomalies might occur in the training set of these datasets as well.

3. The task of obstacle detection is not sufficiently detailed. For example, which pixels are included in the evaluation? In Figure 1, there  is a person walking on the road. It is not labeled as an obstacle, nor is it included in the evaluation mask. Does this mean that if an algorithm wrongly detects the human as an obstacle, this is not reflected in the evaluation metric? If this is the case, I think this would bias the evaluation result.

**Additional Feedback:**

1. For clarity, it would be useful to add a legend to the image in Appendix F5 that explains the color coded class labels.

2. In equation $(1)$, $A(k)$ denotes all GT pixels of the dataset that are not within the GT region $k$. I believe the denominator of the equation should be $|k \cup \hat{K}(k) - A(k)|$ instead of $|k \cup \hat{K}(k)| - |A(k)|$?

General comment: The paper considers an interesting and important problem for which more high quality research datasets are needed. The paper is well written and many recent methods were benchmarked on the proposed dataset. My main concerns are the low number of dataset images as well as the absence of a dedicated training split. In addition, there are several questions in my review above for which it would be great to have an answer to from the authors. In its current form, the weaknesses of the work outweigh its strengths which concludes my paper rating - marginally below acceptance threshold.

**Documentation:**

The authors mention that other existing datasets exhibit numerous annotation errors which leads to problems when evaluating on these datasets. How was the accuracy of the provided ground truth masks ensured for this new dataset?

**Ethics:**

-

**Relation To Prior Work:**

Prior work is adequatly covered. Compared to other datasets, the proposed one does not rely on artificially generated anomalies. However, it does not contain significantly more images than existing datasets.

**Summary And Contributions:**

The paper presents a new dataset and benchmark for the localization of anomalies in autonomous driving scenarios. The dataset considers two different tasks, i.e. anomaly segmentation and road obstacle segmentation. The test splits for each of the tasks contain 100 and 327 images, respectively. Pixel precise ground truth annotations are provided for each of the test images. Images for training are not provided. An extensive evaluation of existing methods is performed using pixel-level and component-level metrics. Results indicate room for improvement for future methods when trained on the Cityscapes dataset.

---

> ### Author Response · Authors · 2021-09-27
> **Reply to Reviewer j9s3**
>
> Thanks for this comprehensive review.
>
> > The number of provided images is small
>
> While it is correct that the number of images is smaller, in both our datasets, RoadAnomly21 as well as RoadObstalce21, all images differ considerably. We added example images from other similar datasets in Figure 17, where the images do not differ noticeably.
>
> The similarity of images in other datasets can also be derived from Table 1. For the anomaly track, our dataset contains 100 images, each in a different scene. In comparison, Fishyscapes LostAndFound has 373 images recorded in 12 different environments. Furthermore, there are only 7 different anomaly types. Thus, there are at most 12*7=84 different combinations of anomaly and environment, which means 84 different scenes. Moreover, our dataset contains significantly more anomaly pixels (0.23% vs ours: 13.83% of the respective datasets). The latter also holds when comparing against the CAOS Anomaly dataset (0.83% anomaly pixels).
>
> For the obstacle track, other datasets achieve their high number of images by densely sampling from video sequences. For instance, LostAndFound contains 1203 images from 13 environments and 9 obstacle categories. Since some scenes contain more than one object, usually 10 or more frames are taken from the same scene (environment-obstacle combinations), depicting anomalies from different distances. RoadObstacle21 provides about 3 images from the same scene, thus arriving at a similar image per scene ratio compared to LostAndFound. However, RoadObstacle21 contains 31 anomalous objects while LostAndFound only contains 9.
>
> We conclude that the overall scale of our dataset therefore is very comparable with other existing datasets in the field. In particular, the number of combinations of scenes and anomalous objects is even higher than in most other datasets.
>
> ---
> > The dataset does not include a training set. The paper also does not contain a clear recommendation on which other datasets should be used for model training.
>
> We explain in line 262 that all of our reference methods are trained on Cityscapes. However, we now understand that the manuscript did not clearly enough state that this is not a formal requirement. We therefore added a paragraph in the manuscript explaining our recommendations.
>
> In general, we intentionally do not provide any anomaly training data since the task of anomaly detection is to identify anything semantically different from the already used training data.
>
> --
> > the task of obstacle detection is not sufficiently detailed. For example, which pixels are included in the evaluation? In Figure 1, there is a person walking on the road. It is not labeled as an obstacle, nor is it included in the evaluation mask
>
> The obstacle track focuses on the task of distinguishing between drivable area (road) and non-drivable area (obstacle), as defined in Section 3.1. The goal is to make sure that the road ahead of the ego-car is free of any hazard, irrespective of the object category of potential obstacles. The mentioned image from Figure 1 (with a person walking on the road) is an example of LostAndFound. It illustrates the inconsistent labeling of LostAndFound which we state in our manuscript. The inconsistent labeling prevents a proper evaluation of anomaly/obstacle segmentation methods, which is one reason why we introduced our RoadObstacle21 dataset. Despite more diversity of images, our dataset also has a higher label quality.
>
> ---
> >Does this mean that if a small prediction is fully within a large GT component, this prediction will be marked as a FP since its sIoU with the GT is quite small?
>
> Thanks for noticing this issue. It is correct that sIoU<threshold would mean that a small prediction fully within a large ground truth component will be marked as false positive (FP), and we agree that this definition needs to be modified. We changed the definition of false positive components by considering the PPV (positive predictive value or equivalently component-wise precision) instead of the sIoU. In other words, predicted components are marked as false positives if their PPV<threshold. With the PPV, only the portion of incorrectly predicted pixels of a predicted component determine whether the entire component is considered as FP, irrespective of the (size of the) intersecting ground truth components.
>
> We changed the definition in the manuscript, rerun all evaluations and updated all tables.
>
> Moreover, the adaptation using PPV is only applied to the definition of FP. For the definitions of TP and FN, we stick to the proposed sIoU.
>
> ---
> > How was the accuracy of the provided ground truth masks ensured for this new dataset?
>
> We are a team of people from three different working groups. All images and corresponding annotations were checked several times by multiple people. As a quality assessment, each labeled image was reviewed by at least two further team members. We will state this in the labeling policy section in the updated manuscript.

---

> > ### Author Response · Authors · 2021-09-27
> > **Further reply to Reviewer j9s3 (regarding training data)**
> >
> > > For example, if the training dataset does not cover a wide range of domains, an algorithm trained on this dataset could rightfully mark areas as anomalous that are labeled as anomaly-free
> >
> > We apologize for the confusion caused by our insufficient definition of anomaly. Given our refined definition of anomaly, which we explain in our general comment on domain shift (see top), the problem given in the cited example cannot occur. It is never seen as correct to label a pixel of a known semantic class as anomaly, regardless of how big the domain shift may be. We therefore differentiate between semantic anomalies and visual novelties, where the latter should not be segmented. Most of the methods that we tested, however, are not trained on segmentation itself, but output pixel-wise scores. In these cases, our performance metrics neither punish nor reward if visual novelties get high anomaly scores (due to the use void class), but simply require that semantic anomalies get assigned higher scores than on known semantics.
> >
> > In our experiments we already see that it depends on the method whether very diverse training data is required or not. All our evaluated models use Cityscapes data, which shares the same setup as the LostAndFound dataset (same lighting, camera, location, ...). We report results on LostAndFound for comparison purposes in Table 4 and have a respective leaderboard on the website. Comparing the performance of the same set of methods on LostAndFound and RoadObstacle21 allows us to evaluate the impact of domain shift, for which we observe that it indeed can affect some of the tested methods. In RoadObstacle21, we include various road surfaces and a variety of obstacle types. The performance of methods that are highly based on statistics of Cityscapes, such as Mahalanobis or Embedding density, therefore suffer drastically. However, other methods specifically designed for anomaly segmentation, such as Road inpainting, SynBoost or maximized Entropy, show to be considerably more robust against the slightly changing environments.
> >
> > In our experiments, we used the DeeplabV3+ model as the underlying semantic segmentation model, which we also recommend to use and which is trained on ImageNet, Mapillary and Cityscapes. It is hard to draw a line between pretraining feature encoders (e.g. on ImageNet), training the semantic classes (what we do on Cityscapes), and specific supervision of the anomaly detection (which we allow and track as ‘OoD data’ in the leaderboards). The challenge of using OoD data is the choice of the right proxy such that the trained anomaly detector generalizes to our wide variety of anomaly objects. But, if necessary, any other additional data is also allowed to be used to better cope with the domain of our datasets. The latter is again another reason why we do not intend to publish a dedicated training set.
> >
> > > Since objects were defined as anomalous based on the class definitions of the Cityscapes dataset, it seems that using datasets that localize different classes would not be suitable since possible anomalies might occur in the training set of these datasets as well.
> >
> > We choose Cityscapes definitions because they represent a standard policy for semantic segmentation datasets in automated driving. The class definitions of Cityscapes were also used in e.g. the DarkZurich dataset [A], the India Driving Dataset [B] (where more classes are annotated, but annotators were specifically calibrated to Cityscapes definitions), or the GTA-5 dataset [C] and there exist compatible mappings to e.g. BDD-100k [D] (the authors evaluate Cityscapes-trained models).
> > However, it is true that some methods may train on other data that includes parts that we consider as anomalous. In some cases, like the methods SynBoost, VoidClassifier, or Maximized Entropy, this is intended as they specifically use auxialariy datasets to train anomaly detection in a supervised way. In other cases, it may indeed unintentionally harm methods as what we annotate as anomaly may be known to a model. We therefore clarified our recommendation for training data on lines 263-268. Given the high compatibility of the Cityscapes label definitions, we argue that there is still a huge body of data available to any anomaly detection method. Extending training data could however also potentially harm and not help methods, which is why we leave it at this recommendation. Submissions can freely tailor their training regime and the benchmark will reveal what regimes work and what not.
> >
> > ---
> > [A] C.Sakaridis et al; Map-Guided Curriculum Domain Adaptation and Uncertainty-Aware Evaluation for Semantic Nighttime Image Segmentation, T-PAMI 2020
> >
> > [B] G Varma et al - IDD: A Dataset for Exploring Problems of Autonomous Navigation in Unconstrained Environments - WACV 2019
> >
> > [C] S.Richter et al; Playing for Data: Ground Truth from Computer Games; ECCV 2016
> >
> > [D] F.Yu et al; BDD100K: A Diverse Driving Dataset for Heterogeneous Multitask Learning, CVPR 2020

---

> > > ### Comment · Reviewer_j9s3 · 2021-09-30
> > > **Thank you for your response. Some concerns remain.**
> > >
> > > Thank you for taking the time to respond and the improvement of
> > > the evaluation metric, which is fine with me. Taking the entire rebuttal into
> > > account, my main concerns remain:
> > >
> > > 1. Number of test samples:
> > >
> > > I agree that the scale of the dataset is comparable to LostAndFound
> > > and CAOS. However, I still believe that the number of test images
> > > should be increased, especially for the anomaly track.
> > > For a safety critical task such as autonomous driving,
> > > assessing the performance on only 100 images seems problematic to me
> > > given the abundance of possible anomalies that may occur in practice.
> > > This is independent of existing datasets, for which I also believe that
> > > the number of test images is insufficient.
> > >
> > > 2. Domain shift:
> > >
> > > I still think it is problematic that the majority of test samples differ
> > > significantly from the recommended training set Cityscapes. I understand
> > > that the goal of the authors is to test the robustness of anomaly detectors
> > > under some domain shift. However, browsing through the 100 test images
> > > of the anomaly track, I find the domain shift with respect to the Cityscapes
> > > training set to be extreme for many of the test samples. For example:
> > >
> > > - Numerous images show close-up views of animals where the street /
> > >   surrounding environment is barely visible in the image
> > >   (pig0000, pig0001, bear0000, zebra0000, cow0001, horse0004, elephant0010).
> > > - caravan0007 was apparently taken by a pedestrian from a bridge with a
> > >   significant camera angle and height difference to the road.
> > >
> > > I am concerned that the domain shift introduced by the test set is unnatural
> > > for real-world autonomous driving applications and weakens the reliability of the method ranking
> > > in the presented benchmark.

---

> > > > ### Author Response · Authors · 2021-09-30
> > > > **Reply to Reviewer j9s3**
> > > >
> > > > Thank you very much for your comment.
> > > >
> > > > Regarding the mentioned images we have the corresponding score maps here:
> > > > * https://github.com/SegmentMeIfYouCan/road-anomaly-benchmark/tree/master/doc/RoadAnomalyExamples/Entropy_max.
> > > >
> > > > The anomaly score maps are produced by the maximized entropy method, which also achieves a state-of-the-art mIoU score of 89.3% on the semantic segmentation of the Cityscapes Validation dataset (cf table 5). We observe that methods are already capable to deal with anomalies under the present domain shift in RoadAnomaly21. However, there is still much room for improvement. This is why, the road anomaly track should also challenge researchers to develop new methods, to show which training regimes could work in anomaly segmentation and which do not. Moreover, a certain amount of domain shift is intentional since it represents situations in driving perception, in which no training data was captured.

---

### Official Review · Reviewer_dqy5 · 2021-09-20
**A useful dataset for anomaly and obstacle detection for autonomous vehicle settings**

**Rating:** 6
**Confidence:** 3
**Correctness:** The dataset curation, experimental se…
**Clarity:** The paper is well-written.

**Strengths:**

1.	The paper tackles anomaly detection, a critical problem in ML evaluation, particularly in real-world settings like autonomous driving.
2.	The experimentation and evaluation is very well detailed and explains how anomaly detection can be explored in a cohesive evaluation environment
3.	The authors have provided very detailed documentation (including a website), a leaderboard, and evaluation suite to facilitate interaction with the data


**Weaknesses:**

1.	The provided datasets classify anomalies/obstacles based on classes in only one dataset – CityScapes. Different AV and semantic segmentation datasets may have different labels, so something that is an anomaly relative CityScapes may not be an anomaly relative to another dataset. Do the authors have some guidelines for submitting models to this challenge? Do models have to be trained on the CityScapes dataset?
2.	The authors hide the labelled test data from all users, which may limit user’s ability to rapidly prototype models and access data. The authors do provide a limited number of images and annotations for each dataset, which will help future users, but it would be helpful if the authors can detail how they anticipate users to interact with the dataset especially when users need to rapidly benchmark their models (i.e. for deadlines, etc).
3.	The authors mention that RoadObstacle21 contains 85 images in night and in extreme weather. It would be helpful if the authors provide benchmark results on this dataset, even though it has a distribution shift, to showcase how current out of the box methods perform on this data.


**Additional Feedback:**

N/A

**Documentation:**

The dataset seems to be very well documented. The evaluation suite though was not visible at time of review (link is broken).

**Ethics:**

Not that I can tell

**Relation To Prior Work:**

The related work is well detailed and Table 1 gives a good summary of the differences between this dataset and other datasets.

**Summary And Contributions:**

This paper introduces two datasets for anomaly and obstacle detection in addition to a well-established evaluation suite for benchmarking models against these datasets. Additionally, the work provides baseline networks from different family of approaches for anomaly detection and explores the impact of different evaluation criteria on perceived performance.

---

> ### Author Response · Authors · 2021-09-27
> **Reply to Reviewer dqy5**
>
> > Different AV and semantic segmentation datasets may have different labels, so something that is an anomaly relative CityScapes may not be an anomaly relative to another dataset.
>
> Thank you for pointing out the importance of the labeling policy. We choose Cityscapes definitions because they represent a standard policy for semantic segmentation datasets in automated driving. The class definitions of Cityscapes were also used in e.g. the DarkZurich dataset [A], the India Driving Dataset [B] (where more classes are annotated, but annotators were specifically calibrated to Cityscapes definitions), or the GTA-5 dataset [C] and there exist compatible mappings to e.g. BDD-100k [D] (the authors evaluate Cityscapes-trained models).
>
> Moreover, even datasets for automated driving, that have incompatible labels, should have in common that the driveable area is distinguishable from other classes. Our obstacle track focuses on the task of distinguishing between drivable area (road) and non-drivable area (obstacle). Here, the obstacles do not necessarily have to be semantic anomalies (except for not fitting into the definition of drivable area). The motivation of this track is to make sure that the drivable area is free of any hazard. Therefore, our labeling policy is not in conflict with the labeling policy of other datasets for automated driving, but rather ensures high compatibility.
>
> ---
>
> [A] C.Sakaridis et al; Map-Guided Curriculum Domain Adaptation and Uncertainty-Aware Evaluation for Semantic Nighttime Image Segmentation, T-PAMI 2020
>
> [B] G Varma et al - IDD: A Dataset for Exploring Problems of Autonomous Navigation in Unconstrained Environments -  WACV 2019
>
> [C] S.Richter et al; Playing for Data: Ground Truth from Computer Games; ECCV 2016
>
> [D] F.Yu et al; BDD100K: A Diverse Driving Dataset for Heterogeneous Multitask Learning, CVPR 2020
>
> ---
> >Do the authors have some guidelines for submitting models to this challenge?
>
> The choice of which data to use to train their methods is up to the participants in the benchmark. In our experiments, we used the DeeplabV3+ model as the underlying semantic segmentation model, which we also recommend to use and which is trained on ImageNet, Mapillary and Cityscapes. The “OoD data” columns in our results tables indicate whether additional data was used to train the anomaly detector. Here, the challenge is to find the right proxy such that the anomaly detector generalizes to our wide variety of anomaly objects. Any other data is allowed to be used for that, and in our experiments the COCO dataset has proven to be a good proxy (see method maximized entropy).
>
> There are two reasons that kept us from restricting the choice of which data to use to train the their methods:
> * It is impossible to enforce any training regime except for participants submitting training code, data, and network structures which is beyond the support capacity that we can provide as independent research labs.
> * It is hard to draw a line between pretraining feature encoders (e.g. on ImageNet), training the semantic classes (what we do on Cityscapes), and specific supervision of the anomaly detection (which we allow and track as ‘OoD data’ in the leaderboards).
>
> ---
> > The authors hide the labelled test data from all users, which may limit user’s ability to rapidly prototype models and access data
>
> We appreciate and understand the concern for the speed of our research area. However, publishing the labels of the test data would extremely lower the credibility of our public leaderboard. To prototype methods, our evaluation-suite can be used for both, our small validation sets and test set images. The code will not output the performance metrics if there is no ground truth available. However, our code provides tools for visualization to at least visually check how well methods perform.
>
> Moreover, we are a team of more than four people who maintain the benchmark. Our latest submission was processed within a day. The participants only have to make sure to properly prepare their method using our code example (https://github.com/SegmentMeIfYouCan/road-anomaly-benchmark/blob/master/example_inference.py) and send us the generated outputs. A routine (also included in our publicly available evaluation-suite) then further processes the outputs on our local machines and computes performance metrics, which are sent back to the participants.
>
> Other datasets, such as LostAndFound, are publicly available including their labels and can be used for rapid prototyping. What is rather missing in this research area is a credible leaderboard that uses real-world data without participants being able to overfit their methods on publicly available labels.
>
> ---
> > RoadObstacle21 contains 85 images in night and in extreme weather. It would be helpful if the authors provide benchmark results on this dataset
>
> We thank the reviewer for their suggestion. We added the benchmark results for the night and snowstorm scenes in Table 12.

---

> > ### Comment · Reviewer_dqy5 · 2021-10-01
> > **Response to authors**
> >
> > The authors' responses are appreciated. I have provided detailed answers to the points below
> >
> > 1. **Different AV and ...** It would be helpful to clarify this in the text. Currently, the impression I received as a reader was that the labeling policy for the Cityscapes dataset must be followed for all other datasets on which models could be trained
> > 2. **Do the authors ...** A section in the text or appendix that mentions this "Rules for Usage" would be helpful for future users of the dataset
> > 3. **The authors hide ...** While I understand the decision to keep the dataset closed, this decision does impact the speed of the iteration cycle for different methods. Other datasets like ImageNet where the test set is hidden have large enough validation sets on which most benchmarking is done. Such a subset of data that is accepted and established as a benchmarkable subset would be very useful to the community in which prototyping is rapid and constantly evolving.
> > 4. Thank you for providing this table

---

### Official Review · Reviewer_EzQc · 2021-09-21
**Review of SegmentMeIfYouCan**

**Rating:** 7
**Confidence:** 3
**Correctness:** Yes
**Clarity:** Yes

**Strengths:**

1. The motivation of an anomaly segmentation dataset is good with real-world applications.
2. The proposed dataset is well constructed and documented.
4. A very comprehensive benchmark is included containing results from many different methods. Detailed settings are also provided.

**Weaknesses:**

1. Although the anomaly(obstacle) categories are more diverse, the overall scale of the dataset is still relatively small compared to other similar datasets.
2. This dataset doesn't provide a training set while the domain shift between the training and test set could greatly affect the models' performance. Since all models are trained on Cityscapes, this paper doesn't show how different training domains may affect models' performance.

**Additional Feedback:**

None

**Documentation:**

Yes

**Ethics:**

No ethics issues

**Relation To Prior Work:**

Yes

**Summary And Contributions:**

This paper introduces an anomaly(obstacle) segmentation dataset with real-world applications. A detailed benchmark is also included.

---

> ### Author Response · Authors · 2021-09-27
> **Reply to Reviewer EzQc**
>
> > the overall scale of the dataset is still relatively small compared to other similar datasets
>
> While it is correct that the number of images is smaller, in both our datasets, RoadAnomly21 as well as RoadObstalce21, all images differ considerably. We added example images from other similar datasets in Figure 17, where the images do not differ noticeably.
>
> The similarity of images in other datasets can also be derived from Table 1. For the anomaly track, our dataset contains 100 images, each in a different scene. In comparison, Fishyscapes LostAndFound has 373 images recorded in 12 different environments. Furthermore, there are only 7 different anomaly types. Thus, there are at most 12*7=84 different combinations of anomaly and environment, which means 84 different scenes. Moreover, our dataset contains significantly more anomaly pixels (0.23% vs ours: 13.83% of the respective datasets). The latter also holds when comparing against the CAOS Anomaly dataset (0.83% anomaly pixels).
>
> For the obstacle track, other datasets achieve their high number of images by densely sampling from video sequences. For instance, LostAndFound contains 1203 images from 13 environments and 9 obstacle categories. Since some scenes contain more than one object, usually 10 or more frames are taken from the same scene (environment-obstacle combinations), depicting anomalies from different distances. RoadObstacle21 provides about 3 images from the same scene, thus arriving at a similar image per scene ratio compared to LostAndFound. However, RoadObstacle21 contains 31 anomalous objects while LostAndFound only contains 9.
>
> We conclude that the overall scale of our dataset therefore is very comparable with other existing datasets in the field. In particular, the number of combinations of scenes and anomalous objects is higher than in most other datasets.
>
> ___
>
> > This dataset doesn't provide a training set while the domain shift between the training and test set could greatly affect the models' performance.
>
> We intentionally do not provide any anomaly training data since the task of anomaly detection is to identify anything semantically different from the already used training data. Methods could overfit on anomaly training data, which is particularly undesirable in anomaly detection.
>
> Since the definition of the domain of anomalies depends on the training domain, we will more precisely state that we expect the Cityscapes-classes to be treated as non-anomalies. Note that we do state this in line 187-189 of the manuscript. A training set would only ease this task of anomaly detection and limit the generalization statements that we ultimately seek.
>
> ___
> > Since all models are trained on Cityscapes, this paper doesn't show how different training domains may affect models' performance.
>
>
> Regarding the domain shift between the semantic segmentation training and our dataset, we refer to our specific comment to all reviewers above.
>
> All our evaluated models use Cityscapes data, which shares the same setup as the LostAndFound data (same lighting, camera, location, ...). We report results on LostAndFound for comparison purposes in Table 4 and have a respective leaderboard on the website. Comparing the performance of the same set of methods on LostAndFound and RoadObstacle21 allows us to evaluate the impact of domain shift, for which we observe that it indeed can affect some of the tested methods. In RoadObstacle21, we include various road surfaces and a variety of obstacle types. The performance of methods that are highly based on statistics of Cityscapes, such as Mahalanobis or Embedding density, therefore suffer drastically. However, other methods specifically designed for anomaly segmentation, such as Road inpainting, SynBoost or maximized Entropy, show to be considerably more robust against the slightly changing environments.
> There are two reasons that kept us from restricting the choice of which data to use to train the methods:
> * It is impossible to enforce any training regime except for participants submitting training code, data, and network structures which is beyond the support capacity that we can provide as independent research labs.
> * It is hard to draw a line between pretraining feature encoders (e.g. on ImageNet), training the semantic classes (what we do on Cityscapes), and specific supervision of the anomaly detection (which we allow and track as ‘OoD data’ in the leaderboards).
>
> In our experiments, we used the DeeplabV3+ model as the underlying semantic segmentation model, which we also recommend to use and which is trained on ImageNet, Mapillary and Cityscapes. The challenge of using OoD data is the choice of the right proxy such that the trained anomaly detector generalizes to our wide variety of anomaly objects. But, if necessary, any other additional data is also allowed to be used to better cope with the domain of our datasets. The latter is again another reason why we do not intend to publish a dedicated training set.

---

> > ### Comment · Reviewer_EzQc · 2021-09-29
> > **Reply to authors**
> >
> > Thank you very much for the detailed response. I have raised my score accordingly.

---

### Author Response · Authors · 2021-09-27
**General Answer Regarding Domain Shift and Definition of Anomalies**

To properly evaluate anomaly segmentation, some domain shift is necessary. In this sense, we are thankful that the reviews pointed us to improve our definition of anomalies, as we now do in lines 140-146. If our data had no domain shift, it would be impossible to differentiate between a method that performs good semantic anomaly segmentation, i.e. segmenting only what does not belong into the class definitions, and a novelty segmentation method that segments out anything that appears visually different to the training data, such as a car in a novel color. If a method is very vulnerable to domain shift, it is by our definition therefore not a good anomaly segmentation method and consequently receives low scores in our benchmark.

We however also acknowledge that class definitions can be fuzzy (lines 194-196), which is why we
* use the ‘void/ignore’ label to exclude semantically ambiguous parts from the evaluation (e.g. rickshaws), and
* designed RoadObstacle21 to focus on the more clear and practically relevant drivable areas and obstacles.

While the definition of anomaly in our benchmark closely matches with the CAOS and Fishyscapes benchmarks, we understand that the term ‘anomaly’ is also used in other works e.g. for production anomalies or visual outliers and it may have been unclear to readers and reviewers without the revised definition what the purpose of our benchmark is.

---

### Decision · Program_Chairs · 2021-10-09

**Decision:**

Accept

**Comment:**

Although reviewers agree on the high relevance of the problem, there are common critiques regarding missing justifications/details, limited data size, and issues related to the lack of a training set. After rebuttal several of the critiques of the reviewers have been carefully discussed and addressed by the authors. Overall the paper achieves the minimum score to be accepted for publication at the NeurIPS data track.